# Towards Interactive Global Geolocation Assistant

**Zhiyang Dou**[*] **Zipeng Wang**[*] **Xumeng Han**[*] **Guorong Li** **Zhenjun Han**[†]

University of Chinese Academy of Sciences
hanzhj@ucas.ac.cn

## Abstract

Global geolocation, the task of predicting the exact location of street-view images, is crucial for applications like security surveillance. Existing retrieval and classification methods, along with current Multimodal Large Language Models (MLLMs), suffer from limitations such as database dependence, lack of interpretability, and a significant gap in geographic knowledge due to insufficient datasets. To address these issues, we introduce `MG-Geo`, the first comprehensive, high-quality **M**ulti-modal **G**lobal **Geo**location dataset. Comprising five million instances of geographic dialogue data across 210 countries, `MG-Geo` provides detailed geographic element cues (*e.g.*, road markings, vegetation, language), significantly surpassing existing datasets like OSV-5M and Google Landmark V2 in richness and granularity. Leveraging `MG-Geo`, we develop **GaGA** (**G**lobal **G**eo-location **A**ssistant), a novel MLLM specifically designed for geolocation. Experimental results demonstrate that GaGA not only significantly outperforms existing MLLMs but also surpasses the state-of-the-art model OSV-5M-Baseline in administrative boundary prediction (achieving improvements of 4.57% at the country and 2.92% at the city levels). Furthermore, GaGA exhibits remarkable interactive refinement capabilities, improving localization accuracy with effective user guidance. This work highlights the critical role of the `MG-Geo` dataset in fostering improved geographic understanding of MLLM. Our dataset is accessible via: https://huggingface.co/datasets/kendouvg/MG-Geo.

## 1 Introduction

Global geolocation aims to predict the exact location of any street-view image, with wide applications in security surveillance, emergency response, disease outbreak prediction, environmental monitoring, and tourism navigation [49, 50, 37, 40]. This task requires integrating visual cues, such as road signs, architectural styles, climate, and vegetation, with geographic knowledge to accurately predict GPS coordinates or location labels. For images with landmarks or distinctive architecture, the location can be inferred by combining visual features with contextual knowledge. However, geolocation becomes more challenging in homogenous environments, such as highways or natural landscapes, where subtle geographic clues like road markings, license plate types, and signage must be relied upon.

The existing street-view localization methods are generally categorized into retrieval-based and classification-based approaches. The retrieval-based methods [48, 51] match input images with similar ones from a geotagged database but are constrained by the diversity and completeness of the database. The classification-based methods [38, 43] classify images into predefined regions based on visual features, but they lack interpretability and fail to provide explicit visual cues. Additionally, several studies have explored leveraging text content [15, 1, 20, 36, 23] and social

---

[*]Authors contributed equally to this work.
[†]Corresponding author.

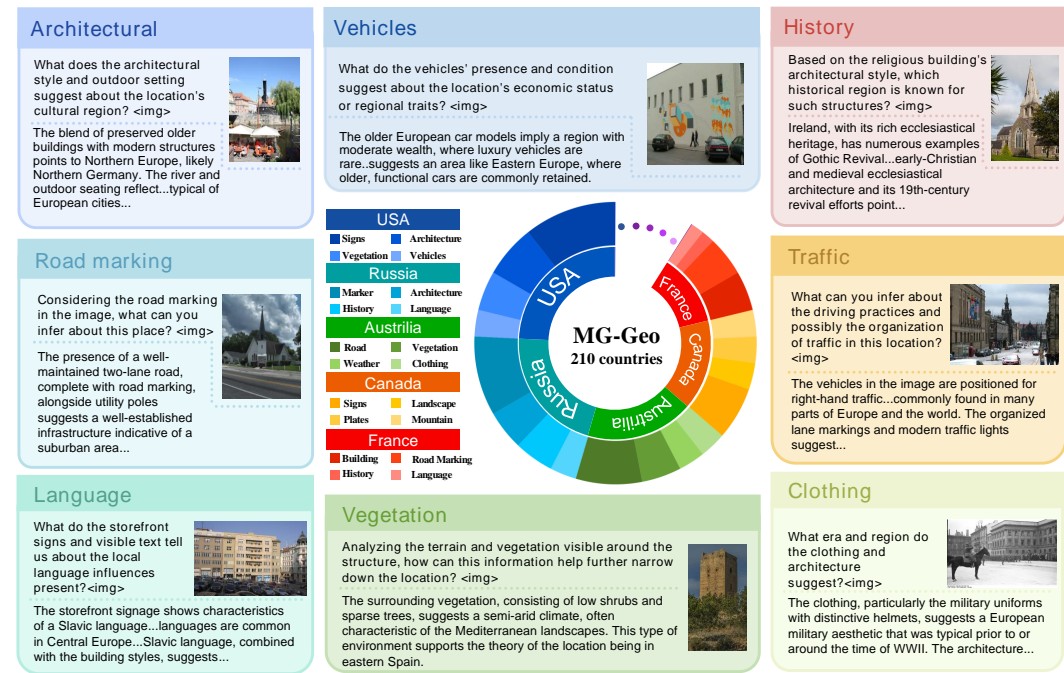

Figure 1: Illustration of `MG-Geo`. Featuring diverse geographic scenes and visual cues, these images demonstrate the utility for training MLLMs to connect visual content with geographic locations and enrich their understanding of global environments.

network relationships [5, 26, 35] for geolocation by analyzing user-generated content and social interactions.

In practical scenarios, geolocation is rarely a one-time, static process; it involves integrating and refining multiple sources of information iteratively through continuous interaction. Traditional geolocation models directly regress geographic labels or coordinates, inherently lacking interpretability and flexibility. Multimodal large language models (MLLMs), such as [29, 9, 32], are renowned for their ability to integrate multimodal information and are capable of using this knowledge for interpretative reasoning, which is especially important for applications like geolocation. However, the existing MLLMs encounter substantial challenges in global geolocation, particularly due to the geographic knowledge gap within their LLMs [37] and inability to establish associations between visual features of geographic elements and their corresponding locations. A primary factor underlying this observation is that existing MLLMs datasets, such as those presented in [7, 17, 27], omit critical granularities including administrative boundaries and precise geographic coordinates.

To address these challenges, we introduce the first **M**ulti-modal **G**lobal **Geo**location (`MG-Geo`) dataset. In contrast to existing geolocation datasets such as OpenStreetView-5M (OSV-5M) [4] and Google Landmark V2 [46], which provide only basic descriptors (*e.g.*, latitude, longitude, country, region, and city) lacking detailed geographic information, `MG-Geo` is a comprehensive, high-quality dataset comprising five million instances of geographic dialogue data. As illustrated in Figure 1, `MG-Geo` comprises a diverse array of geographic element cues, encompassing road markings, vegetation, and language, among others. With content spanning 210 countries, the dataset demonstrates notable richness and high quality.

Leveraging this dataset, we develop the **G**lob**a**l **G**eo-location **A**ssistant (**GaGA**), a novel MLLM designed to overcome the limitations of the geographic localization tasks' poor explainability and low insight. We train GaGA in two phases: In the first phase, we pretrain the projector of an MLLM using a large image-location dataset to inject geographic knowledge to enhance its ability to classify geographic locations. In the second phase, we finetune the model with a curated subset of image-clue and multi-turn QA pairs data to improve the models's capacity for interaction and reasoning. The experimental results demonstrate that GaGA not only significantly outperforms similar MLLMs on the GWS15k dataset but also surpasses the current state-of-the-art model, OSV-5M-Baseline, in

Table 1: **Comparison between `MG-Geo` and existing datasets**. `MG-Geo` is the first large-scale multimodal dataset curated for the domain of geolocation.

| Dataset | Size | Open-access | Source Type | Scope | QA Pairs | Chain of Thought |
|---------|------|-------------|-------------|-------|----------|------------------|
| Im2GPS3k[21] | 3k | ✓ | Web-scraped | Biased | ✗ | ✗ |
| YFCC4K[44] | 4k | ✓ | Web-scraped | Biased | ✗ | ✗ |
| MP-16[42] | 4.7M | ✓ | Web-scraped | Biased | ✗ | ✗ |
| GWS15k[10] | 15k | ✗ | Street-view | Global | ✗ | ✗ |
| OSV-5M[4] | 5M | ✓ | Street-view | Global | ✗ | ✗ |
| Google Landmark V2[46] | 5M | ✓ | Landmark | Global | ✗ | ✗ |
| **MG-Geo (ours)** | 5M | ✓ | Street-view and landmark | Global | ✓ | ✓ |

predicting the administrative boundaries with improvements of 4.57% and 2.92% at the country and city levels, respectively. Notably, GaGA possesses the capability to refine its responses in interactive scenarios. When users provide effective guidance or correct priors, GaGA's localization accuracy improves substantially.

## 2 Related work

### 2.1 Geolocation Datasets

In the domain of geolocation, the localizability of images within datasets is of paramount importance. Though composed of a wealth of geotagged images, the existing datasets, such as Im2GPS3k [21], YFCC4K [44] and MP-16 [42], though composed of a wealth of geotagged images, contain many unlocatable images and exhibit distribution biases. GWS15k [10] mitigates distribution differences, and ensures that the images are authentic, localizable street views; however, this dataset is not open-source. OSV-5M [4] is the largest open-source collection of planet-scale, localizable street view images. The Google Landmark V2 [46] dataset contains globally distributed human-made and natural landmarks and showcase iconic landscapes.

We propose `MG-Geo`, the first multimodal geolocation dataset designed to enhance the perception and interactivity of MLLMs in geolocation tasks. Curated from OSV-5M and Google Landmark V2, `MG-Geo` offers a clean, evenly distributed resource. It also incorporates well-structured global language knowledge, providing a dataset that better reflects the complexity and diversity of real-world geolocation challenges.

### 2.2 Geolocation Models

Mainstream geolocation methods can be broadly categorized into two approaches: image-based retrieval and classification-based methods. Image-to-image retrieval techniques rely on dense image retrieval libraries, which perform well for localization tasks within small areas. However the cost of constructing such retrieval libraries on a global scale is prohibitively high. When geolocation is treated as a classification task, categories can be defined based on administrative regions, divided into geocells according to specific rules, or discretized into latitude and longitude coordinates. TransLocator [47] employs images and semantic segmentation maps as inputs, facilitating interaction between two parallel branches after each Transformer layer and enabling multitask geolocation and scene recognition. GeoCLIP [43] introduces a location encoder and applies random Fourier feature representations to latitude and longitude coordinates. It utilizes the pretrained CLIP [34] visual encoder to represent images and aligns them with the corresponding location features for localization. Pigeon [19] is a method that classifies within self-created geocell and retrieves locations within clusters.

In recent years, some works have begun to explore the potential of natural language in geolocation tasks. G3 [30] predicts the country of an image by automatically extracting clues from human-written guidebooks. StreetCLIP [18] employs captions containing geolocation information for contrastive learning, allowing the use of natural language to ground CLIP in the context of image geolocalization. GeoReasoner [28] is the first work to fine-tune a Multimodal Large Language Model (MLLM) for street view image localization. Unlike GeoReasoner, our model is trained on a planet scale and multimodal dataset of localizable images, which is not confined to narrow distributions and proposes an interactive approach to accomplish the localization task.

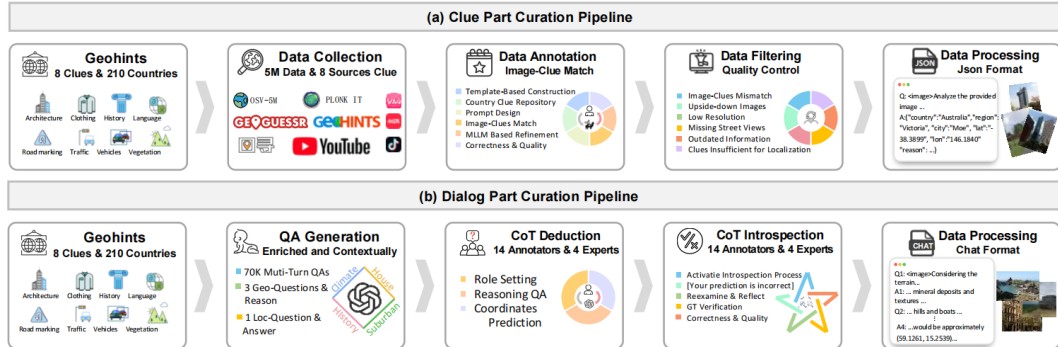

Figure 2: **An illustration of our pipeline for data curation.** (a) We construct the Clue Part by leveraging guidance clues from online geolocation game communities and employing an MLLM. (b) We generate location-agnostic, multi-turn reasoning QA pairs and high-quality dialog data for the Dialog Part, applying the Interactive Reasoning CoT method to activate CoT Deduction and CoT Introspection tasks.

## 3 MG-Geo Dataset

In this paper, we introduce `MG-Geo`, a novel dataset encompassing a diverse array of geographic element cues, including architecture, environment, landmarks, and climate across various countries. The dataset is structured into three distinct components: the *Meta Part*, the *Clue Part*, and the *Dialog Part*, designed to accommodate disparate training objectives. We leverage structured geographic knowledge, elicited from expert GeoGuessr players, and human-guided interactions with powerful MLLMs to facilitate the construction of this dataset. `MG-Geo` not only addresses the existing gap in geographic knowledge within LLMs but also enhances their perception of geographic cues, enabling interpretable reasoning for geolocation prediction. Furthermore, the improved geographic understanding fostered by this dataset has the potential to pave the way for future research in applications such as navigation and place retrieval.

### 3.1 Meta Part

In the Meta Part, images and meta-geographic information are taken from the OSV-5M [3], which inherits its characteristics of good distribution, wide scope, and high quality. After removing a small number of samples with incomplete location annotations, we organize each sample into JSON format using three levels of administrative boundaries—country, region, and city. This results in a total of 4.87 million entries, covering 70k cities, 2.7k regions, and 210 countries.

### 3.2 Clue Part

We design our model to generate textual clues from geographical features in images to enhance output interpretability. Users can assess and correct these clues during interaction and provide additional information to improve the model's accuracy. We design an automatic multimodal QA generation paradigm to convert source-cued annotations into different forms of QAs. Figure 2(a) shows an automated pipeline for generating high-correlated image-text clues pairs.

#### 3.2.1 MLLM Based Refinement

Note that although the 3,000 guidance clues crawled from the GeoGuessr game and Tuxun game manual contain rich geographic localization information, directly inputting these text-based clues into GaGA for learning may not allow it to fully utilize the data. It is because pure text lacks the supporting image features that provide the necessary contextual information for the reasoning process. To overcome this issue, we leverage MLLMs' multimodal input advantage. MLLMs excel at processing image and text data by matching each clue with its corresponding image representation, empowering GaGA and enhancing its reasoning ability with the clues.

To ensure the image representations' general applicability across various contexts, we follow the sampling method in [10] and select 70k globally distributed samples from the OSV-5M dataset

for clue matching. We divide the clue matching process into two main steps: *constructing of the country-specific clue repository* and *matching of image-clue pairs.*

The construction of the country-specific clue repository is a manual classification process in which, each clue is categorized based on its associated country or region, ensuring that each country/region has a set of specific clues (*e.g.*, *United States: [Clue 1], [Clue 2], . . . ,[Clue N].*) On this basis, matching image-clue pairs involves associating street-view images with specific clues from the country-specific repository to generate image-text clue pairs. We use a MLLM [8], which generates natural language descriptions (*), and its corresponding geographic clues (*e.g.*, `<image>[Clue 1, Clue 3, Clue M]`*.) for each image. The core of this process is to guide the model in selecting and summarizing geographic clues that are helpful for location identification. The process is complete only when the selected clues are validated by the recognizable features within the image, ensuring that the final output contains accurate geographic clues for each sample.

### 3.2.2 Human verficication

Some ambiguity (*e.g.*, *upside-down images, low resolution, or missing street views*) and errors are still inevitable despite the use of manually annotated data sources, clues from the geolocation game, and carefully designed quality assurance methods. During the *Clue Part* construction process, we implement a manual validation protocol: when evaluators flag ambiguous or erroneous image-clue pairs, we trace the source of the errors and either remove problematic data samples or modify the metadata accordingly to adjust, the image or clue descriptions. This manual validation step ensures that the natural language descriptions of the clues accurately correspond to the intended target.

## 3.3 Dialog Part

As shown in Figure 2(b), we begin the *Dialog Part* construction process by standardizing a well-annotated subset of the Google Landmark V2 into a unified metadata structure, ensuring the generation of multi-turn reasoning QA pairs that are location-agnostic. In order to enhance GaGA's reasoning depth and conversational ability by supporting the analysis of images from multiple perspectives and inferring specific locations, we select 73K samples from Google Landmark V2 with rich information such as architecture, vegetation, cultural elements, and climate. Then, with the assistance of GPT-4V, we generate QA pairs using the Interactive Reasoning CoT method.

### 3.3.1 Question-Answer Generation

We intend to create image descriptions that thoroughly capture visible appearance and attributes, integrating relevant knowledge, climatic characteristics, architectural styles, and even historical context. This all-encompassing strategy ensures the dataset's robust support for a broad spectrum of real-world applications by providing enriched and contextually rich data. For example, an image of a typical suburban house in Chicago might reveal the following features: **Cold Climate**: *A steep gable roof, designed to handle snowfall reflects the typical cold climate typical of the northern regions of North America;* **Distinct Seasons***. The use of stone, wood...*

The generation of multi-turn QAs mainly relies on providing unified metadata and carefully designed prompts to MLLMs, specifically GPT-4V. Through this process, GPT-4V engages in multi-turn self-questioning based on the image, gradually guiding the model to reason through and uncover the geographical information. Each set of multi-turn QAs includes the following key attributes: *question ID, source dataset, image path, three geo-questions w/ reasoning process, and one loc-question w/ ultimate answer*. This structure ensures the logical coherence of the multi-turn QAs and clearly presents the progression from question to reasoning process to the final answer.

Prompting techniques improve LLMs' reasoning and problem solving abilities across diverse tasks [22, 24, 41, 45]. We integrate the images' unified metadata format to generate high-quality dialog data. Using the Interactive Reasoning CoT method, we activate two tasks: *CoT Deduction* and *CoT Introspection*. In the next part, we elaborate on the implementation details of these two tasks.

### 3.3.2 CoT Deduction

To extract the reasoning chain behind the geographic location predictions from GPT-4V as the training data, we explicitly extract the reasoning chain supporting the model's QA process. Specifically,

we draw on the concept of interactive reasoning from reinforcement learning and propose the *CoT Deduction* method to handle the geographic location prediction task.

*The CoT Deduction* consists of three parts: *Role Setting, Reasoning QA, and Coordinate Prediction*.

- *Role Setting*. In CoT Deduction, we set up two roles: *Geo-Guessr player* and *questioner*. The questioner and player interact, with the questioner asking questions and the player responding based on the image clues and existing knowledge. The interactive reasoning model in reinforcement learning allows the model to interact with the environment, continuously adjusting the reasoning process through repeated trials and feedback. Thus, the questioner and player jointly advance the reasoning process in CoT Deduction.

- *Reasoning QA*. We aim to explicitly extract the internal principles of geographic location reasoning to construct `MG-Geo`'s Dialog Part. For each question from the questioner, the Geo-Guessr player gradually deduces the geographic location based on various aspects embedded in the image, such as the environment and climate, architecture and landmarks, language and culture, and people's appearance. Each QA round (*i.e.*, Q1A1, Q2A2, and Q3A3) helps the player narrow down the possibilities, gradually approaching the correct answer.

- *Coordinate Prediction*. After a series of reasoning steps, the player needs to provide a specific geographic coordinate and briefly explain their choice (Q4A4).

During the process, the temperature and GPT-4V's top-p and top-k parameters are set to 1, 1, and NONE, respectively, to ensure the stability and accuracy of the generation process. After *CoT Deduction* generates the predicted coordinates, we initiate a *Decision Criterion* to evaluate the predicted coordinates' accuracy. Specifically, we calculate the Haversine distance between the predicted coordinates and the unified metadata. If the distance between the predicted and true coordinates is greater than $25km$, the *CoT Introspection* process is triggered.

### 3.3.3 CoT Introspection

After activating the *CoT Introspection* process, we input the prompt *[Your prediction is incorrect]* and provide the actual geographic coordinates and corresponding location as a reference. It encourages GPT-4V to reexamine the image and reflect on the reasoning generated during the *CoT Deduction* process. Meanwhile, the model must identify and correct any errors in the reasoning, as well as fill in any key information and clues that are previously overlooked.

The purpose of providing the real coordinates is to ensure that the reflection process leads to more accurate and reliable reasoning. It is important to note that the model parameters, question setup, and dialog structure during the *CoT Introspection* process remain consistent with those of the *CoT Deduction* process, ensuring that the reflection results can seamlessly replace the incorrect answers from *CoT Deduction* to generate a complete and correct reasoning dataset.

## 4 GaGA

Capitalizing on the introduced `MG-Geo` dataset, we present a novel MLLM termed GaGA. In contrast to the prevalent *"black box"* nature of existing geolocation models that yield predictions devoid of explanatory context, GaGA integrates robust geolocation capabilities with the capacity to associate and leverage extensive world knowledge, thereby enabling dynamic and context-aware predictions during user interaction. Specifically, when a user queries a geographic feature or provides pertinent prior information, GaGA can effectively fuse this input with its internal knowledge base to generate more informed and nuanced predictions.

### 4.1 Model Setting

GaGA uses the same model architecture and training objectives as LLaVA [29], which consists of a vision encoder $f_{VM}$ for extracting features $f_v$ from street view images, a projector layer $f_P$ for feature mapping, a Large Language Model (LLM) $f_L$ ,such as Llama3 [2], and a text tokenizer $f_T$. We select the pretrained Llama3-8B as $f_L$ because it excels in mapping coordinates to geographic names among publicly available LLMs. Implementation details can be found in the Appendix.

## 4.2 Training Framework

The training process of GaGA is divided into two distinct stages: pretraining and finetuning, each with specific objectives and methodologies designed to progressively refine the model's capabilities.

**Pretraining.** The primary objective during the pretraining phase is to enable the model to develop a basic and intuitive understanding of images from a variety of regions. At this stage, the vision encoder and LLM parameters remain fixed, and only the projector's parameters are updated. We train the model, using data from the Meta Part of `MG-Geo`, which contains diverse image-text pairs that cover a broad range of geographic contexts.

**Finetuning.** Following pretraining, the finetuning stage focuses on adapting the model to effectively analyze geographical images and engage in interactive dialogues with users, which is critical for specialized tasks in GaGA. The projector's parameters are fixed, which ensures that the model does not deviate from the fundamental visual understanding it has developed. Instead, the focus shifts to finetuning the LLM to enhance its ability to interpret and interact with the geographical content. The finetuning dataset is a combination of carefully curated subsets of three parts of `MG-Geo`. These datasets provide a comprehensive training foundation for the model's specialized capabilities. The final finetuning dataset consists of 240k image-text pairs, ensuring a diverse and well-rounded input for the LLM adaptation.

# 5 Expriment

To demonstrate the efficacy of our dataset in addressing the geographic knowledge gap in existing models and to showcase its potential in downstream geolocation tasks, we conducted a comprehensive suite of experiments, the primary findings of which are presented in this section. Numerous additional experiments and further details are provided in the Appendix for thoroughness.

## 5.1 Experimental Setup

**Benchmark.** GWS15k is a high-quality benchmark with well-distributed global coverage. However, since it is not an open source, we have reproduced it in this study. We use the test set of OSV-5M as the database and collect evenly distributed imagery based on 43K cities and the surface area of each country. The pseudocode is shown in appendx.

**Metrics.** We employ three metrics to evaluate the geolocation model's prediction accuracy:

- Accuracy of predicted geographical names at various administrative levels: country, region and city.
- Accuracy of predicted coordinates within various distance thresholds: 1km, 25km, 200km, 750km, and 2500km, calculated as the haversine distance between the model's predicted GPS coordinates and the ground truth.
- Geoscore: it is defined as $5000exp(-\delta/1492.7)$ based on the famous Geo-Guessr game. $\delta$ represents the Haversine distance between predicted and ground truth image locations.

**Evaluation Mode.** We employ two evaluation modes, **hierarchical (HIER)** and **direct (DIRE)**. The "HIER" mode is primarily applied in the following scenario: for MLLMs that have not been finetuned on `MG-Geo`, we provide candidate administrative boundary names at each level to constrain their representation of administrative boundaries. In "DIRE" mode, the model directly predicts the location without constraints or hierarchical guidance. In Tables 5.2, 8 and 9, we use the "DIRE" mode as the default setting.

## 5.2 Geolocation Performance

The results of the administrative boundary prediction accuracy are shown in the left side of Table 5.2. To be clear, there is no guarantee that MLLM-based methods will consistently provide relevant answers. Therefore, we use recall rates to measure the proportion of valid answers in a large language model. GaGA demonstrates outstanding performance, surpassing the current state-of-the-art model—OSV-5M-Baseline—with a lead of 4.57% at the country level and 2.92% at the city level. It also achieves performance comparable to the best-performing models at the region level. Additionally,

Table 2: **Administrative-Level Accuracy and Coordinates Accuracy of GaGA and Open-Source Models on GWS15K Bench. Left:** Administrative-Level. [†] indicates MLLM with comparable parameter counts. We use **bold** to indicate the best performance, '___' for the second-best, and '~~~' for the third-best, respectively. **Right:** Coordinates Accuracy. ★ represents the model evaluated on GWS15k reproduced in this paper.

| Method | Evaluation Mode | Recall | Admin-Level Accuracy | | |
|---|---|---|---|---|---|
| | | | Country | Region | City |
| LLaVA-Llama3[†] | HIER | 0.99 | 1.76 | 0.26 | 0.02 |
| InternVL2[†] | HIER | 0.96 | 24.74 | 4.20 | 0.48 |
| Qwen-VL[†] | HIER | 0.98 | 34.20 | 8.19 | 1.45 |
| GeoReasoner[†] | HIER | 1 | 40.63 | 9.57 | 1.11 |
| StreetCLIP | HIER | 1 | 40.11 | 10.75 | 3.02 |
| OSV-5M-Baseline | DIRE | 1 | 58.49 | 29.58 | 3.36 |
| GaGA[†] | DIRE | 1 | 63.06 | 27.95 | 6.28 |

| Method | Coordinates Accuracy (% @ km) | | | | | Geoscore |
|---|---|---|---|---|---|---|
| | 1km | 25km | 200km | 750km | 2500km | |
| ISNs | 0.05 | 0.6 | 4.2 | 15.5 | 38.5 | - |
| Translocator | 0.5 | 1.1 | 8 | 25.5 | 48.3 | - |
| GeoDecoder | 0.7 | 1.5 | 8.7 | 26.9 | 50.5 | - |
| GeoCLIP★ | 0.2 | 3.1 | 15.4 | 40.3 | 71.2 | 2345.2 |
| PIGEON | 0.7 | 9.2 | 31.2 | 65.7 | 85.1 | - |
| OSV-5M-Baseline★ | 0.08 | 14.9 | 39.3 | 56.2 | 74.4 | 2944.9 |
| GaGA★ | 0.1 | 8.5 | 33.9 | 60.6 | 82.2 | 3113.0 |

we compare GaGA with advanced MLLM, such as LLaVA-Llama3, Qwen-VL [6], InternVL2 [8] and GeoReasoner [28]. LLaVA-Llama3 serves as our baseline model, which adopts the LLaVA [29] architecture with Llama3 [2] as its language backbone. Due to the limited size of its train set, its performance on geolocation is significantly poor. For GeoReasoner, we use the Clue Part (73k samples) of MG-Geo and the SFT data (2k samples)[3] provided by the GeoReasoner's authors for "Reasoning Tuning", along with 100k samples from the Meta Part of MG-Geo for "Location Tuning". The results show that GaGA outperforms these state-of-the-art MLLMs in terms of location accuracy. GaGA also outperforms StreetCLIP [18], a model based on the CLIP architecture and finetuned on street-view text data, on the GWS15k dataset.

The Right side of Table 5.2 presents the performance comparison of GaGA with ISNs [31], Translocator [47], GeoDecoder [33], GeoCLIP [43], PIGEON [19], and OSV-5M-Baseline. GaGA performs relatively well in geolocation prediction, achieving the second-best performance across the 200km to 2500km threshold range and the third-best performance at 25km. We evaluate the performance of OSV-5M-Baseline and GeoCLIP on the GWS15k dataset as reproduced in this work to provide a fairer comparison. GaGA outperforms OSV-5M-Baseline at the 1km, 750km, and 2500km granularities, and significantly outperforms GeoCLIP across the 25km to 2500km range. It is worth to notify that GaGA achieves the highest Geoscore among the three models, a metric that strikes a balance by rewarding precise predictions while mitigating the impact of large but infrequent errors. Since the remaining works cannot be reproduced, we are unable to obtain the corresponding Geoscore for them.

Regarding output mechanisms, conventional LLMs exhibit inherent limitations in predicting long floating-point numbers like latitude and longitude [39, 25]. The sequential nature of next-token prediction often necessitates splitting these numbers into multiple tokens, potentially degrading the precision of the resulting floating-point value. Consequently, GaGA's performance at the 1km and 25km thresholds underperforms expectations, underscoring the necessity for improvements in processing high-precision numerical outputs.

## 5.3 Interactive Geolocation Analysis

To further evaluate GaGA's performance in interactive geolocation tasks, we curated a set of 547 images encompassing both cultural and natural landscapes, each paired with meticulously annotated question-answer dialogues. We ensured that questions are closely related to the visible geographical elements without directly providing visual details to guide only location prediction. For example: *"Considering the architectural design, what region of the world would you think displays such forms and why?"*

Table 3 shows the performance of different MLLMs under various questioning modes, including direct inquiry, providing a guiding question (+Q), and a question-answer pair (+QA). By comparing GaGA's performance with its base model—LLaVA-Llama3, on the one hand, we observe that when a guiding question is provided, GaGA shows greater improvement at the region and city levels, except the country level; when an effective geographic prior information is provided, GaGA exhibits a greater performance increase across all levels. That indicates that GaGA has a distinct advantage in generating geographic knowledge related to the question and understanding and integrating prior content to refine its predictions.On the other hand, the decline in GaGA's country-level localization performance

---

[3]https://github.com/lingli1996/GeoReasoner

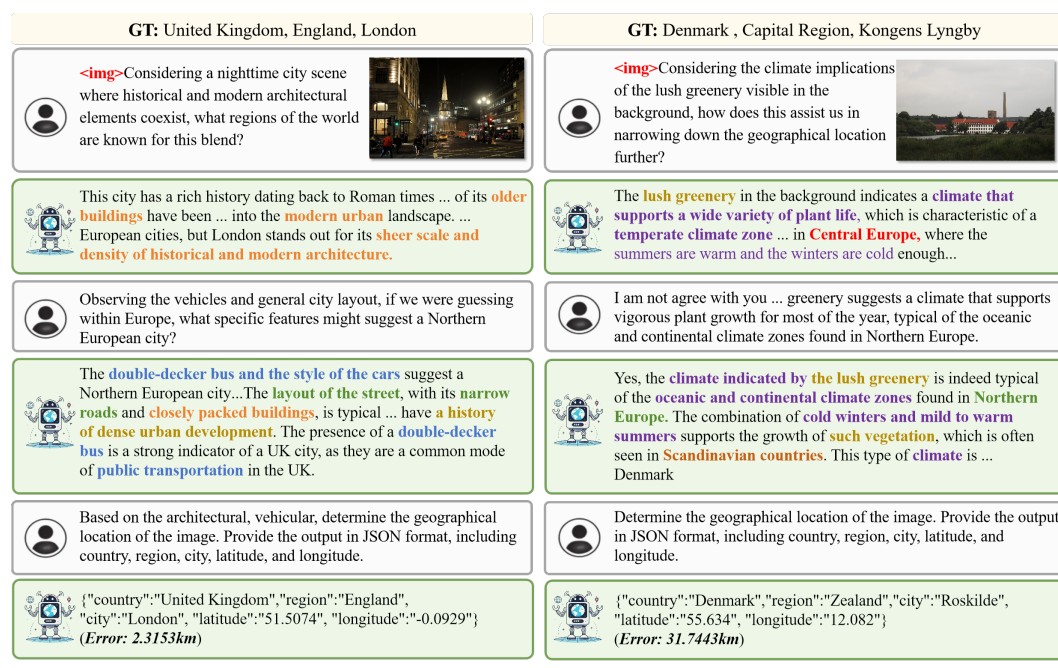

Figure 3: Illustrations of GaGA's dialogues in various scenarios. On the left, we demonstrate how GaGA successfully incorporates external knowledge with human guidance; on the right, we showcase the model's predictive outcomes when given relevant prior information.

under guided questioning is primarily due to the multiple valid responses to geographical feature questions. For example, similar architectural styles across European countries can confuse GaGA at the country level after answering such questions. Conversely, LLaVA-Llama3, with inherently lower country recognition accuracy, benefits from external knowledge, improving the performance by roughly adjusting the prediction range. Additionally, Figure 3 illustrates examples of GaGA's dialogues in different scenarios.

Table 3: Performances of MLLMs with Direct Inquiry, Guiding Question (+ Q), and both question and effective answer (+QA)

| Method | Evaluation Mode | Prompt | Recall | Admin-Level Accuracy | | |
|---|---|---|---|---|---|---|
| | | | | Country | Region | City |
| GaGA | DIRE | Direct inquiry | 1 | 64.89 | 27.97 | 7.67 |
| | | + Q | 1 | 61.24 | 29.25 | 8.22 |
| | | | | -3.65 | +1.28 | +0.55 |
| | | + QA | 1 | 74.77 | 34.73 | 9.87 |
| | | | | +9.88 | +6.76 | +2.2 |
| LLaVA-LlaMA3 | HIER | Direct inquiry | 0.99 | 2.92 | 0.54 | 0 |
| | | + Q | 0.99 | 4.38 | 0.54 | 0.05 |
| | | | | +1.46 | 0 | +0.05 |
| | | + QA | 0.96 | 12.79 | 2.92 | 0.36 |
| | | | | +9.87 | +2.38 | +0.36 |

# 6    Conclusion

In this work, we tackled the challenges in global geolocation, particularly the lack of comprehensive geographic data for MLLMs and the limitations of existing methods. We introduced `MG-Geo`, the first large-scale, high-quality multimodal dataset rich in geographic element cues, specifically designed to bridge the geographic knowledge gap for MLLMs. Leveraging `MG-Geo`, we developed **GaGA**, a novel MLLM demonstrating superior performance over existing models and state-of-the-art baselines in predicting administrative boundaries. Crucially, GaGA's interactive capability allows for refined and more accurate localization based on user input. This research emphasizes the importance of domain-specific high-quality datasets in advancing MLLM capabilities for complex geographical tasks (such as global geolocation), and paves the way for more geographic downstream tasks.

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

## A  Experimental Implementation Details

All experiments are conducted using the XTuner platform [12], facilitating efficient multimodal model tuning and deployment. For reasoning tasks, we employed LMDeploy [11], a toolkit designed for compressing, deploying, and serving LLMs to optimize inference speed and memory efficiency, ensuring real time performance. We conduct all the experimetns are on $8 \times$ RTX4090 GPUs.

**Pretraining.** The projector is initialized using the ShareGPT4V [7] data, which provides pre-existing embeddings that facilitate the mapping of image features to textual descriptions.

**Finetuning.** To optimize the LLM for its task-specific behavior, we apply Quantized Low-Rank Adaptation(QLoRA)[13] to finetune the language model. This technique enables efficient adaptation of the LLM to the specifics of geographical analysis and user interaction without requiring exhaustive retraining of the entire model.

The settings for hyperparameters used throughout the training process include configurations for both pretraining and finetuning stages, along with specifications for the QLoRA and deployment settings. Table 4 and Table 5summarizes the detailed settings we use for pretraining and finetuning. Parameters not mentioned in the finetuning phase are the same as those in the pretraining phase.

<table>
<tr><td colspan="2">Table 4: Pretraining Settings</td></tr>
<tr><th>Configuration</th><th>Value</th></tr>
<tr><td>Dataset</td><td>Meta Part of MG-Geo</td></tr>
<tr><td>Training Epochs</td><td>1</td></tr>
<tr><td>Total Batch Size</td><td>16</td></tr>
<tr><td>Optimizer</td><td>AdamW</td></tr>
<tr><td>LR</td><td>$2\times10^{-4}$</td></tr>
<tr><td>LR Schedule</td><td>CosineAnnealing</td></tr>
<tr><td>Weight Decay</td><td>0</td></tr>
<tr><td>Warmup Ratio</td><td>0.03</td></tr>
<tr><td>Adam Beta1</td><td>0.9</td></tr>
<tr><td>Adam Beta2</td><td>0.999</td></tr>
<tr><td>Image Resolution</td><td>336×336</td></tr>
<tr><td>Max Text Token Length</td><td>1472</td></tr>
</table>

<table>
<tr><td colspan="2">Table 5: Fine-tuning Settings</td></tr>
<tr><th>Configuration</th><th>Value</th></tr>
<tr><td>Dataset</td><td>Mix240k of MG-Geo</td></tr>
<tr><td>Training Epochs</td><td>1</td></tr>
<tr><td>Total Batch Size</td><td>16</td></tr>
<tr><td>Optimizer</td><td>AdamW</td></tr>
<tr><td>LR</td><td>$2\times10^{-5}$</td></tr>
<tr><td>Quantization Type</td><td>BitsAndBytesConfig</td></tr>
<tr><td>Quantization Bits</td><td>4-bit</td></tr>
<tr><td>4-bit Quant Type</td><td>nf</td></tr>
<tr><td>4-bit Compute Dtype</td><td>torch.float16</td></tr>
<tr><td>lora Alpha</td><td>16</td></tr>
<tr><td>Low-Rank Matrix Rank</td><td>64</td></tr>
<tr><td>LoRA Dropout</td><td>0.05</td></tr>
</table>

## B  Model Architecture

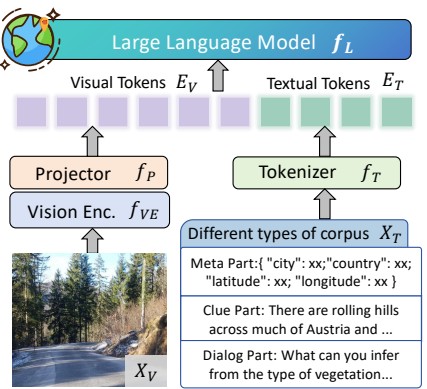

Figure 4: The architecture of GaGA.

As shown in Figure 4, for the input images $X_v$, we employ the pretrained CLIP vision encoder $f_V$, effectively extracting high-level visual features from geographic images. The encoder utilizes the Vision Transformer (ViT) architecture [14], allowing for robust representation of complex visual patterns within the images. Once the visual features are extracted, the projector layer $f_P$ is used to map these representations into the LLM's word embedding space. Specifically, the visual features are encoded into visual tokens $E_V$. The above process is formulated as:

$$E_v = f_P \left( f_{VM} \left( X_v \right) \right) \tag{1}$$

During the training phase, various types of corpus are encoded into textual tokens $E_t = f_T(X_t)$, which are then concatenated with the visual tokens $E_V$. This interaction facilitates a cross-modal exchange between the visual and textual modalities, enabling the model to learn richer, more coherent representations across both domains. Next, all the tokens are fed into the LLM to generate a corresponding output $R$, which is then processed further to produce the final response:

$$R = f_L \left[ E_V, E_T \right] \tag{2}$$

## C  Performances of Advanced MLLMs in Dialog

As shown in the table 6, we evaluate InternVL2 [?] and Qwen-VL [6] in geolocalization under interaction design for performance improvement.Qwen-VL performs poorly under the direct inquiry prompt setting, but its performance at the country level significantly improves after incorporating a guiding question. Similarly, InternVL2, after engaging in dialogue, uncovers more useful clues, leading to performance improvements across the country, region, and city levels, demonstrating the effectiveness of interaction.

Table 6: Performance of advanced MLLMs with different types of prompt inputs.

| Method | Evaluation Mode | Prompt | Recall | Admin-Level Accuracy | | |
|---|---|---|---|---|---|---|
| | | | | Country | Region | City |
| GaGA | DIRE | Direct inquiry | 1 | 64.89 | 27.97 | 7.67 |
| | | + Q | 1 | 61.24 | 29.25 | 8.22 |
| | | | | -3.65 | +1.28 | +0.55 |
| Qwen-VL | HIER | Direct inquiry | 0.96 | 13.89 | 6.03 | 2.01 |
| | | + Q | 0.92 | 21.38 | 6.94 | 1.82 |
| | | | | +7.49 | +0.91 | -0.19 |
| InternVL2 | HIER | Direct inquiry | 0.96 | 54.11 | 19.19 | 3.29 |
| | | + Q | 0.97 | 55.02 | 19.19 | 4.57 |
| | | | | +0.91 | 0 | +1.28 |

## D  Evaluation of Generated Dialogs

As shown in Table 7, we use pairwise ratings (Win, Tie, Lose) against GPT-4V to evaluate GaGA's dialogs on Fluency, Relevance, Informativeness, and Accuracy. "K" represents the Fleiss' Kappa value [16], which is a robust statistical metric that quantifies the degree of agreement among multiple raters who classify items into a fixed set of categories. Three experts have assessed 50 samples and conducted 50 rounds of comparison. In all four evaluation metrics, GaGA consistently outperforms GPT-4V, and the ratings provided by the experts demonstrate a high degree of consistency.

Table 7: Evaluation of GaGA's Dialog on Fluency, Relevance, Informativeness, and Accuracy with Pairwise Ratings Against GPT-4V.

| Metrics | Win | Loss | Tie | K |
|---|---|---|---|---|
| Fluency | 31 | 3 | 16 | 0.55 |
| Relevance | 33 | 5 | 22 | 0.74 |
| Informativeness | 26 | 7 | 17 | 0.64 |
| Accuracy | 22 | 18 | 10 | 0.91 |

## E  Geolocation Performance on Open-Source Bench

Im2GPS3k [44] datasets contain many non-localizable images (*e.g.*, 35% in Im2gps3k lack geolocation), like selfies and indoor photos. Testing on these images could introduce unreliable errors or favor methods that exploit memory training biases in the distribution [3]. For consistency, we report GaGA's performance on and Im2GPS3k, as shown in Table 8. While GaGA achieves a comparable performance to these state-of-the-art models, we believe that the more evenly distributed and challenging GWS15k dataset, as discussed in Section 5.2, provides a more accurate reflection of GaGA's actual localization performance.

## F  Ablation Experiments

In Section 5.2, GWS15k is used as a subset of OSV-5M-test. To address any distribution differences from the sampling strategy, we evaluate the entire OSV-5M test set and report GaGA's performance.

Table 8: Performances on Im2GPS3k Bench.

| Benchmark | Method | Coordinates Accuracy (% @ km) | | | | |
|---|---|---|---|---|---|---|
| | | 1km | 25km | 200km | 750km | 2500km |
| Im2GPS3k | PlaNet | 8.5 | 24.8 | 34.3 | 48.4 | 64.6 |
| | CPlaNet | 10.2 | 26.5 | 34.6 | 48.6 | 64.6 |
| | ISNs | 10.5 | 28.0 | 36.6 | 49.7 | 66.0 |
| | Translocator | 11.8 | 31.1 | 46.7 | 58.9 | 80.1 |
| | GeoDecoder | **12.8** | 33.5 | 45.9 | 61.0 | 76.1 |
| | PIGEON | 11.3 | **36.7** | **53.8** | **72.4** | **85.3** |
| | GaGA | 11.7 | 33.0 | 48.0 | 67.1 | 82.1 |

The entire test set consists of 210,122 images, which are well distributed globally and have excellent diversity. As shown in Table 9, the performance difference between GaGA and OSV-5M-Baseline aligns with Section 5.2's findings. GaGA excels in coordinate prediction accuracy within the 750 km and 2500 km thresholds and leads in administrative boundary classification accuracy at the country and city levels.

Table 9: Comparison of coordinates and administrative-level accuracy between OSV-5M-Baseline and GaGA.

| Model | Coordinates Accuracy | | | | | Admin-Level Accuracy | | |
|---|---|---|---|---|---|---|---|---|
| | 1km | 25km | 200km | 750km | 2500km | Country | Region | City |
| OSV-5M-Baseline | **0.10** | **17.05** | **47.60** | 66.27 | 81.18 | 67.43 | **39.31** | 6.07 |
| GaGA | 0.06 | 8.02 | 40.06 | **67.98** | **85.39** | **71.49** | 37.86 | **7.46** |

Furthermore, as shown in Table 10, we evaluate the impact of the training framework on the GaGA's performance. Since our baseline model—LLaVA-Llama3—cannot produce valid coordinate outputs, the accuracy of coordinate predictions is not reported in this part. It can be observed that after pretraining, the GaGA-pretraining model achieves the highest accuracy in localization, though lacking flexible conversational abilities. The finetuning stage, which incorporates dialog data, slightly reduces localization accuracy but enables the model to flexibly integrate user-provided knowledge and analyze geographical features. Ultimately, we strike a balance between localization performance and conversational ability.

Table 10: Impact of Training Framework on GaGA's performance.

| Method | Evaluation Mode | Recall | Admin-Level Accuracy | | |
|---|---|---|---|---|---|
| | | | Country | Region | City |
| LLaVA-Llama3 | HIER | 0.99 | 1.76 | 0.26 | 0.02 |
| GaGA-pretraining | DIRE | 0.99 | **63.38** | **28.84** | **6.47** |
| GaGA-finetuning | DIRE | 1.00 | 63.06 | 27.95 | 6.28 |

# G   Discussion

Integrating MLLM into image-based geographic localization enhances interpretability, interactivity, and accuracy, benefiting applications like emergency response and environmental monitoring. However, there are still many scenarios in this field that deserve further explorations:

**Failure Cases.** GaGA still faces limitations in distinguishing locations with similar scenes. For instance, when dealing with European countries with similar architectural styles, GaGA may confuse them, as evidenced by the results in Table 3. Furthermore, if users are unable to provide effective guidance, the model's performance can deteriorate. These issues highlight the necessity of further research into knowledge extraction based on MLLMs to achieve more complex geographic localization capabilities. Simultaneously, it is also important to design effective evaluation mechanisms during interactions to retain and update correct information. To improve GaGA's localization accuracy, researcher should focus on enhancing the model's self-correction and adjustment mechanisms to

better adapt to complex and dynamic geographic environments while optimizing localization results through effective user guidance.

**Multimodal Integration for Enhanced Localization.** Looking toward the future, the integration of additional modalities beyond visual and textual data offers the potential to further enrich the representation of geographic images, leading to improved localization performance and interactive capabilities. For example, future research may consider incorporating auditory data, such as ambient sounds from street view images. Similarly, the inclusion of temporal data, such as time-of-day or seasonal variations, could enable the model to interpret geographic images more accurately by recognizing how certain locations change over time. Furthermore, combining data from various sensors, like satellite images, weather patterns, and traffic data, could create a more comprehensive and context-aware system for geographic localization. By incorporating these diverse modalities, MLLMs can improve their ability to discern fine-grained details of a location, facilitating more dynamic and responsive interactions with users.

**Privacy Risks and Responsible Deployment.** The use of MLLMs faces significant ethical challenges, particularly concerning privacy risks associated with sensitive location data. As these models process large volumes of geospatial data, including potentially personal or private information, concerns about user privacy and data security arise, especially if data is collected without explicit consent or shared in violation of privacy regulations. To mitigate these risks, researchers should protect sensitive information, ensure transparency in data usage, and implement safeguards against misuse. Additionally, while MLLMs offer substantial benefits in improving geographic localization, their deployment must be carefully managed. Responsible deployment involves addressing model limitations, managing biases in training data, ensuring transparency in data handling, and prioritizing user privacy. By balancing technological advancement with ethical considerations, MLLMs can serve society effectively while safeguarding stakeholders' rights and interests.

## H Reproduction of Validation Set GWS15k

To collect evenly distributed imagery, we used a database of 43,000 cities and each country's surface area. We first sampled countries/regions based on their proportion of Earth's surface area, then randomly selected a city within each and GPS coordinates within a 5 km radius of that city's center to sample from OSV5M-Test. Figure 5 presents the global distribution of our test dataset, GWS15k. As depicted in the figure, the sampling points are uniformly distributed across the globe. This uniform distribution ensures that our dataset encompasses a wide range of geographical variations, providing a comprehensive basis for the robust evaluation and generalization of our proposed methods. We provide the pseudo-code for the reproduction of GWS15k.

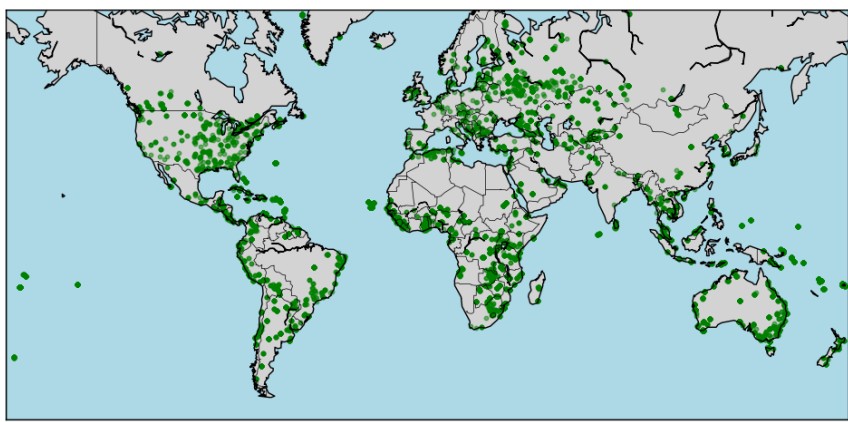

Figure 5: Distribution of GWS15k.

---

**Algorithm 1** Reproduction of GWS15k

---

0: **Input:**
0:   $C$ (Cities dataset), $Co$ (Countries dataset), $Coord$ (GPS coordinates)
0:   $N_{max}$ (Max valid locations), $R$ (Radius: 5 km)
0: **Output:**
0:   $V$ (Valid locations)
0:
0: **function** COMPUTEPROB($Co$, $A_{total}$)
0:   **for** each $c \in Co$ **do**
0:      $P_{base}[c] \leftarrow \frac{Area[c]}{A_{total}}$, $P_{adj}[c] \leftarrow 0.5 \times P_{base}[c] + \frac{0.5}{|Co|}$
0:   **end for**
0:   **return** $P_{adj}$
0: **end function**
0:
0: **function** GENVALIDLOC($C, Co, Coord, P_{adj}, N_{max}, R$)
0:   $V \leftarrow \emptyset$
0:   **while** $|V| < N_{max}$ **do**
0:      Normalize $P_{adj}$
0:      $c_s \leftarrow$ sample from $Co$ with $P_{adj}$
0:      $S \leftarrow \{city \in C \mid city.country = c_s\}$
0:      $s_s \leftarrow$ sample from $S$
0:      $coord_c \leftarrow s_s.coordinates$
0:      **for** each $coord \in Coord$ **do**
0:         $d \leftarrow$ haversine($coord_c$, $coord$)
0:         **if** $d \le R$ **and** $coord \notin V$ **then**
0:            Add $coord$ to $V$
0:         **end if**
0:      **end for**
0:   **end while**
0:   **return** $V$
0: **end function**=0

---

# I  Prompts Employed in the Clue Part Generation

To ensure question variety, we design multiple templates for each question type following the approach outlined in [29]. These templates provide variation while maintaining focus on the geolocalization task. For example, the following are some templates we use in the *Clue Part*:

- ```
  Analyze the given image for clues that help in geolocation and
  combine these clues to localize the image.  Output the answer in
  JSON format.
  ```

- ```
  Can you identify the place where this image was taken?  Analyze
  the street view image from multiple angles to infer its geographic
  location and output the results and clues in JSON format.
  ```

- ```
  Where was this image taken?  Analyze the image in conjunction with
  the geographic clues in the image.  Outputs localization results and
  inference clues in JSON format.
  ```

# J  Prompts Employed in the Dialog Part Generation

The *CoT Deduction* prompt that guides the model through the steps of reasoning and prediction is as follows:

> **CoT Deduction Prompt**
>
> [Role Setting]
> You are an excellent GeoGusser player and questioner. The player deduces the location step by step from clues like environment, climate, buildings, culture, and appearance, while the questioner guides deeper analysis to uncover more clues.
>
> [Reasoning QA]
> 1. Based on the image provided to you; please conduct THREE rounds of QAs (Q1A1, Q2A2, and Q3A3) between the questioner and the player.
> 2. Questions should be sufficiently challenging and closely related to the visual elements but NOT actively provide visual details to the player.
> 3. Only include questions that guide position prediction and require the player to utilize complex reasoning, world knowledge, and interpretive answers to gradually deduce the location. When answering complex questions, provide detailed reasoning steps for clarity and persuasiveness.
>
> [Coordinate Prediction]
> 1. After the reasoning, the questioner should ask about the geographic coordinates and request an answer from the player, denoted as Q4A4.
> 2. Based on previous rationale and analysis, the player makes the best prediction and briefly explains the choice. The player MUST provide reasonable coordinates regardless of uncertainty. Please use Decimal Degrees for coordinates and STRICTLY follow this JSON format: {(latitude, longitude)}

The *CoT Introspection* prompt that encourages GPT-4V to reexamine the image and reflect on the reasoning generated during the *CoT Deduction* process is as follows:

> **CoT Introspection Prompt**
>
> [Attention]
> Your prediction is incorrect!
>
> [Reflecting]
> The actual geographic coordinates are {(X, Y)}. Please revise your answers (A1-A4) base on this. You should correct the wrong deduction and supplement overlooked clues.
>
> [Request]
> 1. Use a reasoning tone.
> 2. The correct coordinates MUST be given in A4.
> 3. Keep the questions (*i.e.*, Q1 to Q4) consistent and include them in your response.
> Please use Decimal Degrees for coordinates and STRICTLY follow this JSON format: {(latitude, longitude)}

## K    Additional Qualitative Results

Additional Qualitative results of GaGA are shown in Figure 6 and Figure 7.

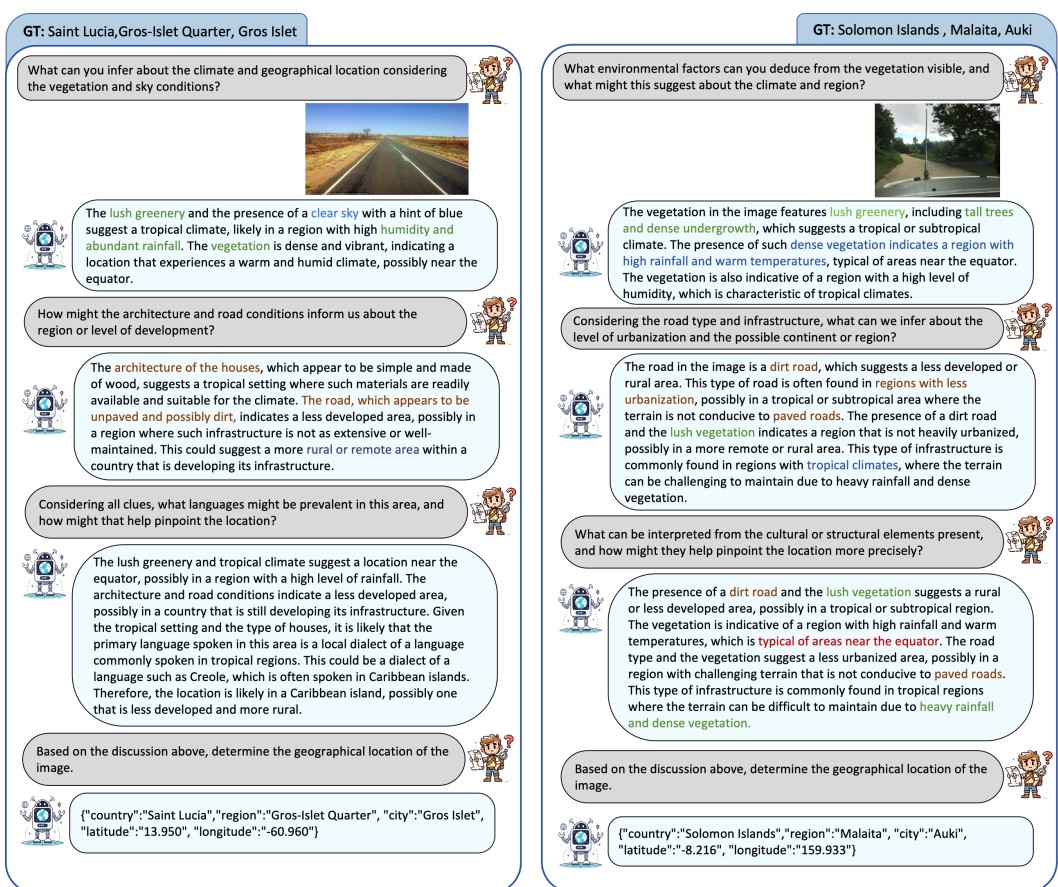

Figure 6: Qualitative results of GaGA on rural images.

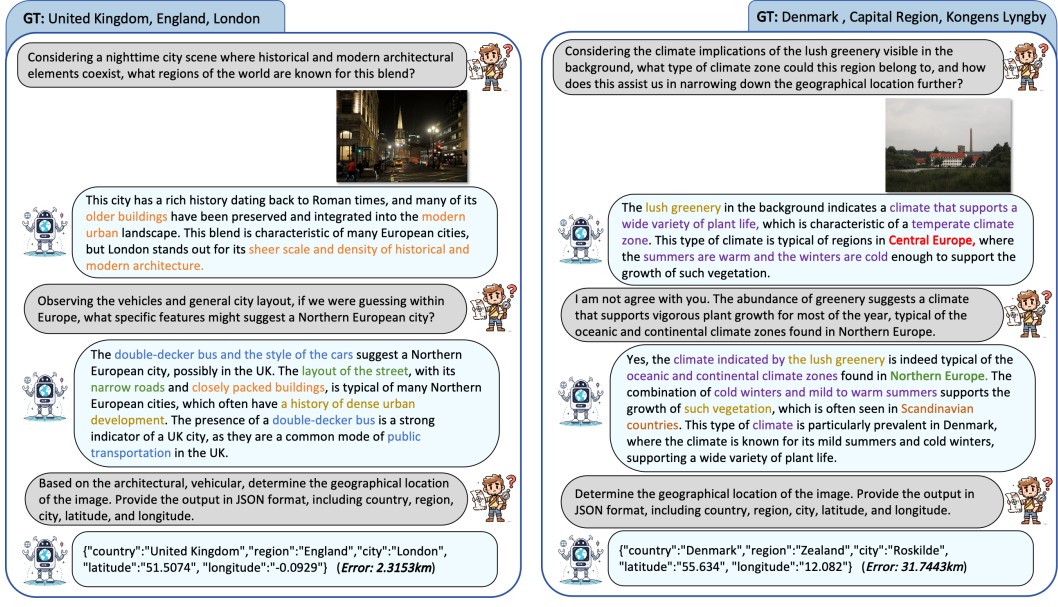

Figure 7: Qualitative results of GaGA on rural images.

