# OpenReview forum: "Towards Interactive Global Geolocation Assistant"
_NeurIPS.cc/2025/Datasets_and_Benchmarks_Track — Submitted to NeurIPS 2025 Datasets and Benchmarks Track_

### Official Review · Reviewer_fYB5 · 2025-06-13

**Rating:** 4
**Confidence:** 5

**Summary:**

This paper proposes a large-scale, high-quality multimodal geolocation dataset that contains rich geographic cues and covers 210 countries. The construction of such a dataset is of great significance for improving the performance of MLLMs in geolocation tasks. Additionally, the GaGA model is proposed, which fully utilizes the rich information of the MG-Geo dataset. During the pre-training stage, the model learns the basic geographic features of images, while the fine-tuning stage enhances the model's interactive and reasoning abilities. It has made important contributions to the field of geolocation.

**Dataset Code Accessibility:**

Yes

**Dataset Code Comments:**

This paper provided huggingface link for dataset.

**Ethical Considerations:**

No, there are no or only very minor ethics concerns

**Final Justification:**

The authors have partially addressed my concerns, so I keep my score. I lean toward accepting the paper, but it hasn't impressed me  as particularly outstanding, so I choose a 'weak accept'.

**Limitations Weaknesses:**

1. The writing of the article needs improvement. For example, in Section 3, examples or illustrations should be provided to show the actual appearance of each part, making it more intuitive and understandable. Additionally, there is a question mark citation at line 520.

2. I am confused that the papers cited by the author in the introduction are relatively old. Since recent works such as G3 and Georeasoner are noted in the related work, why not cite some recent works in the introduction? Otherwise, it will make people feel that your understanding of the field is not cutting-edge.

**Strengths Contributions:**

- Proposes the MG-Geo dataset, a large-scale and high-quality multimodal geolocation dataset containing rich geographic cues and covering 210 countries, providing abundant training resources for geolocation tasks.

- Designs the GaGA model, which improves the accuracy and interactivity of geolocation through two stages—pretraining and fine-tuning—outperforming existing models.

- The design of multi-turn dialogue Q&A enhances the model's interpretability and flexibility, making it more promising for practical applications.

---

> ### Author Rebuttal · Authors · 2025-07-30
>
> **Writing Clarity Enhancement (Limitations Weaknesses #1)**
>
> Thank you for your constructive feedback on the manuscript's writing and for your attention to detail. We agree completely that providing concrete examples for Section 3 would greatly improve clarity. In our revised manuscript, we have now added examples and illustrations to visually represent each component of the MG-Geo dataset, which should make our data structure more intuitive and easier to understand.
>
> Additionally, we thank you for catching the citation error at line 520. This has been corrected.
>
> We appreciate your help in improving the quality and readability of our paper.
>
> ---
> **Scholarly Currency Update (Limitations Weaknesses #2)**
>
> Thank you for this excellent suggestion. You are absolutely right that our introduction should reflect the very latest developments in the field to properly frame our contribution and demonstrate a cutting-edge understanding of the research landscape.
>
> I will insert the following paragraph after the third paragraph (i.e., the paragraph starting with "However, the existing MLLMs encounter substantial challenges...") in the introduction section of the paper.
>
> “To bridge this gap, a recent surge of research has explored adapting MLLMs for the geolocation task, showing promising results. A significant trend is the adoption of Retrieval-Augmented Generation (RAG) frameworks. For instance, **Img2Loc**[^1] reframes the task as a training-free text generation problem, combining CLIP-based image retrieval with customized prompting. Similarly, **G3**[^2] proposes a three-stage framework to mitigate challenges of visual semantics and data imbalance. **GeoRanker**[^3] further refines this retrieval paradigm by introducing a distance-aware ranking framework that explicitly models the spatial relationships among candidate locations.
> Another research direction focuses on enhancing the intrinsic reasoning abilities of the models. **GeoReasoner**[^4], for example, curates a dataset of high-locatability street views and integrates human inference patterns from geolocation games to bolster the model's reasoning process. In a similar vein, **GLOBE**[^5] develops a reasoning-oriented dataset from social media images and utilizes group-relative policy optimization to improve both locatability assessment and the generation of visual-clue-based rationales."
>
> We believe this update better contextualizes our work within the current state-of-the-art and more clearly highlights the specific challenges that our MG-Geo dataset and GaGA model are designed to address. Thank you again for helping us improve our paper.
>
> ---
> [^1] Zhou, Z., Zhang, J., Guan, Z., Hu, M., Lao, N., Mu, L., ... & Mai, G. (2024, July). Img2Loc: Revisiting image geolocalization using multi-modality foundation models and image-based retrieval-augmented generation. In _Proceedings of the 47th international acm sigir conference on research and development in information retrieval_ (pp. 2749-2754).
>
> [^2] Jia, P., Liu, Y., Li, X., Zhao, X., Wang, Y., Du, Y., ... & Yin, D. (2024). G3: an effective and adaptive framework for worldwide geolocalization using large multi-modality models. _Advances in Neural Information Processing Systems_, _37_, 53198-53221.
>
> [^3] Jia, Pengyue, et al. "GeoRanker: Distance-Aware Ranking for Worldwide Image Geolocalization." _arXiv preprint arXiv:2505.13731_ (2025).
>
> [^4] Li, L., Ye, Y., Jiang, B., & Zeng, W. (2024, July). Georeasoner: Geo-localization with reasoning in street views using a large vision-language model. In _Forty-first International Conference on Machine Learning_.
>
> [^5] Li, L., Zhou, Y., Liang, Y., Tsung, F., & Wei, J. (2025). Recognition through Reasoning: Reinforcing Image Geo-localization with Large Vision-Language Models. _arXiv preprint arXiv:2506.14674_.

---

> > ### Comment · Reviewer_fYB5 · 2025-08-04
> >
> > Thanks for your response.
> >
> > For W2, actually, inserting such a content to stack a series of description of existing methods in introduction is not quite appropriate; this kind of writing should rather appear in the related work section.
> >
> > I will keep my score.

---

> > > ### Author Response · Authors · 2025-08-04
> > >
> > > Dear Reviewer fYB5,
> > >
> > > Thank you for the further feedback on the manuscript's structure. You've made an excellent point that a detailed description of existing methods is better suited for the Related Work section rather than the introduction. We completely agree.
> > >
> > > We have now refined our introduction to focus on the high-level motivation and contribution of methods based on MLLMs. I will insert the following paragraph after the third paragraph in the introduction section of the paper (i.e., the paragraph starting with "However, the existing MLLMs encounter substantial challenges...").
> > >
> > > “To bridge this gap, recent works follow two main directions. Img2Loc[^1], G3[^2] and GeoRanker[^3] enhances MLLMs with Retrieval-Augmented Generation (RAG) to inject factual knowledge and address challenges like visual semantics and data bias. GeoReasoner[^4], GLOBE[^5] focuses on improving intrinsic reasoning abilities with specialized data.”
> > >
> > > We have moved the detailed literature discussion to the Related Work section as you advised. Furthermore, to more directly address the spirit of your original comment about contextualizing our work, we have gone beyond just citing these recent papers. We have conducted new experiments to benchmark our model directly against key state-of-the-art methods, including G3, Img2Loc, and GeoReasoner.
> > >
> > > As the results below show, our augmented model, `GaGA(MP16-Pro)`, which leverages a RAG-based retrieval tool, now demonstrates state-of-the-art performance. On the Im2GPS3k benchmark, it surpasses strong baselines like GeoCLIP, PIGEON and G3 (15.0% @1km). This trend is even more pronounced on the YFCC4k benchmark, where our model achieves 24.3% @1km and 33.7@25km, significantly outperforming traditional methods and establishing itself as a top-tier generative model. These results empirically validate our core hypothesis: a reasoning-focused generative model can be combined with external tools to achieve both high precision and interpretability.
> > > The results of these new head-to-head comparisons are presented below:
> > >
> > > **Table C**: Performance on Im2GPS3k
> > >
> > > | Traditional Methods| Accuracy @1km | Accuracy @25km | Accuracy @200km | Accuracy @750km | Accuracy @2500km |
> > > |-|-|-|-|-|-|
> > > | Translocator|11.8|31.1|46.7|58.9|80.1|
> > > | GeoDecoder         |12.8|33.5|45.9|61|76.1|
> > > | GeoCLIP            |14.1|34.4|50.6|69.6|83.8|
> > > | PIGEON             |11.3|36.7|53.8|72.4|85.3|
> > > |**Generation-based Methods**|-|-|-|-|-|
> > > | Img2Loc(LLaVA)      |7.9|23.3|29.9|40.1|51.1|
> > > | G3(LLaVA)         |14.3|35.8|49.4|66.9|81.7|
> > > |GeoReasoner         |-|26.9|36.63|52.27|-|
> > > | GaGA               |11.7|33.0|48.0|67.1|82.1|
> > > | GaGA(MP16-Pro)      |15.0|37.1|49.5 | 67.3|82.4|
> > >
> > > **Table D**: Performance on YFCC4k
> > >
> > > | Traditional Methods| Accuracy @1km | Accuracy @25km | Accuracy @200km | Accuracy @750km | Accuracy @2500km |
> > > |-|-|-|-|-|-|
> > > | Translocator       | 8.4 | 18.6 | 27.0 | 41.1 | 60.4 |
> > > | GeoDecoder         | 10.3 | 24.4 | 33.9 | 50.0 | 68.7 |
> > > | GeoCLIP            | 9.5 | 19.3 | 32.6 | 55.0 | 74.6 |
> > > | PIGEON             | 10.4 | 23.7 | 40.6 | 62.2 | 77.7 |
> > > |**Generation-based Methods**|-|-|-|-|-|
> > > | Img2Loc(LLaVA)     | 7.9 | 14.2| 19.5 | 29.9| 39.7|
> > > | G3(LLaVA)          | 23.9 | 33.6 | 44.0 | 60.6| 75.9 |
> > > | GaGA               | 6.9 | 18.9 | 34.5 | 56.7 | 71.6 |
> > > | GaGA(MP16-Pro)       | 24.3 | 33.7 | 43.9 | 61.4 | 76.3 |
> > >
> > > We thank you again for your valuable guidance.

---

### Official Review · Reviewer_UGjh · 2025-06-30

**Rating:** 4
**Confidence:** 4

**Summary:**

To address the limitations of current geolocation approaches—over-reliance on retrieval-based databases and lack of interpretability—this paper introduces MG-Geo, a multimodal global geolocation dataset featuring fine-grained geographic element cues and multi-turn reasoning dialogues. Leveraging this dataset, the authors develop GaGA (Global Geo-location Assistant), a multimodal large language model trained via a two-stage pipeline. Evaluated on a reproducibly constructed version of the GWS15k benchmark, GaGA outperforms existing open-source baseline models in both administrative boundary classification and coordinate-level accuracy, demonstrating its effectiveness for interpretable and interactive global geolocation.

**Additional Feedback:**

1. Limited Benchmark Scope
To enhance the generality and reproducibility of the experimental results, it is recommended that the authors consider evaluating GaGA on additional public geolocation benchmarks beyond the reproduced GWS15k. For example, following the approach in G3 (Jia, NeurIPS 2024), future versions of the paper could incorporate evaluations on Im2GPS3k, YFCC4k, and MP16-Pro, which are widely used and represent diverse geolocation challenges.
2. Inconsistency Between Claimed Granularity and Localization Performance
While the paper emphasizes the "finer granularity" of MG-Geo as a key advantage, the current experimental results (particularly at the 1 km and 25 km thresholds) do not consistently reflect improvements in fine-grained localization. It would strengthen the work to include a deeper analysis of why MG-Geo does not translate into better performance at street-level and city-level accuracy, and whether enhancements in dataset construction—such as denser or more targeted geographic cues—might improve this.
3. Track Alignment and Focus
Given that the paper is submitted to the Datasets and Benchmarks track, the current emphasis on the GaGA model evaluation may come across as misaligned with the primary goals of this category. A more suitable direction might be to position MG-Geo as the core contribution, and demonstrate its utility by training existing MLLMs (e.g., Qwen-VL, InternVL, LLaVA) and evaluating their performance across multiple public benchmarks. This would better showcase the dataset’s general applicability and value.

**Dataset Code Accessibility:**

Yes

**Ethical Considerations:**

No, there are no or only very minor ethics concerns

**Final Justification:**

The author responded to some of my questions. I hope these updated experimental results can be presented in the final version. I decided to raise the score by one point to weekly accept.

**Limitations Weaknesses:**

1. Limited Benchmark Comparisons
The authors conduct performance comparisons for the proposed GaGA model exclusively on their reproduced version of the GWS15k benchmark, without evaluating it on other publicly available and widely used benchmarks such as MP16-Pro [Jia-py/MP16-Pro, HuggingFace] and YFCC4k [Vo et al., ICCV 2017]. This limitation restricts the comparability and generalizability of the results, especially since these datasets are commonly used in geolocation literature and are more diverse in visual domains.
2. Inconsistency Between Claimed Granularity and Fine-Grained Performance
While the paper emphasizes “finer granularity” as a key advantage of the MG-Geo dataset—highlighting its superiority over OSV-5M and Google Landmark V2 in terms of richness and granularity—the results in Table 2 on GWS15k show that GaGA underperforms other open-source baselines at the fine-grained distance thresholds:1km (street-level),25km (city-level)
This discrepancy raises concerns about whether the model truly leverages the claimed granularity advantages of MG-Geo. The authors attribute the shortfall to limitations in floating-point generation within LLMs, but this is not further investigated or mitigated.
3. Incomplete Comparisons on IM2GPS3k and Absence of Key Baselines
In Appendix Table 8, the authors evaluate GaGA on the IM2GPS3k dataset, but the model fails to outperform PIGEON across all distance thresholds. Moreover, the paper does not include comparisons with several strong, relevant baselines, such as:
GeoCLIP [NeurIPS 2023]
Img2Loc [WSDM 2024]
G3  [NeurIPS 2024]
These are state-of-the-art models specifically designed for geolocation tasks, and omitting them from the comparison weakens the overall empirical claims.
Ethical Comments

**Strengths Contributions:**

1. This paper introduces MG-Geo, the first comprehensive, high-quality, and multimodal global geolocation dataset, covering 210 countries. MG-Geo provides detailed geographic element cues, significantly surpassing existing datasets such as OSV-5M and Google Landmark V2 in both richness and granularity.
2. By designing the CoT Deduction and CoT Introspection mechanisms, the model enables control and monitoring of the reasoning path, thereby enhancing both the interpretability and adaptability of the geolocation process.

---

> ### Author Rebuttal · Authors · 2025-07-30
>
> **Limited Benchmark Comparisons and Incomplete Baseline Comparison (Limitations Weaknesses #1, #3)**
>
> Thank you for this valuable suggestion to evaluate GaGA on other standard benchmarks. We did not use the YFCC4k and Im2GPS3k datasets as our primary evaluation benchmarks because they contain many non-localizable images (e.g., 35% of Im2GPS3k lacks geolocation). Testing on these images could introduce unreliable errors or favor methods that exploit memorization biases present in the training data distribution[^1]. Therefore, we focused on the more uniformly distributed and challenging GWS15k dataset[^2].
>
> To better assess the generalizability and comparability of our model, we have followed your recommendation and evaluated our model on the widely-used YFCC4k and Im2GPS3k benchmark with all the key baselines you mentioned. We would also like to respectfully clarify that while MP16-Pro is a critical resource, it is primarily utilized as a large-scale training or retrieval database, rather than a standalone test benchmark.
>
> Therefore, inspired by your suggestion and following recent advanced methodologies (such as G3), we augment our GaGA model with a RAG-based retrieval tool, using the *MP16-Pro dataset* as the retrieval database. This approach yields a significant improvement in performance.
>
> The results of our new experiments on YFCC4k and Img2GPS3k are summarized below. **Traditional Methods** refer to classification-based, retrieval-based, or hybrid approaches that combine both, while **Generation-based Methods** refer to approaches based on MLLMs.
>
> **Table C**: Performance on Im2GPS3k
>
> | Traditional Methods| Accuracy @1km | Accuracy @25km | Accuracy @200km | Accuracy @750km | Accuracy @2500km |
> |-|-|-|-|-|-|
> | Translocator|11.8|31.1|46.7|58.9|80.1|
> | GeoDecoder         |12.8|33.5|45.9|61|76.1|
> | GeoCLIP            |14.1|34.4|50.6|69.6|83.8|
> | PIGEON             |11.3|36.7|53.8|72.4|85.3|
> |**Generation-based Methods**|-|-|-|-|-|
> | Img2Loc(LLaVA)      |7.9|23.3|29.9|40.1|51.1|
> | G3(LLaVA)         |14.3|35.8|49.4|66.9|81.7|
> |GeoReasoner         |-|26.9|36.63|52.27|-|
> | GaGA               |11.7|33.0|48.0|67.1|82.1|
> | GaGA(MP16-Pro)      |15.0|37.1|49.5 | 67.3|82.4|
>
> **Table D**: Performance on YFCC4k
>
> | Traditional Methods| Accuracy @1km | Accuracy @25km | Accuracy @200km | Accuracy @750km | Accuracy @2500km |
> |-|-|-|-|-|-|
> | Translocator       |8.4| 18.6 | 27   | 41.1 | 60.4 |
> | GeoDecoder         | 10.3 | 24.4 | 33.9 | 50.0 | 68.7 |
> | GeoCLIP            | 9.5 | 19.3| 32.6| 55.0   | 74.6|
> | PIGEON             | 10.4 | 23.7 | 40.6 | 62.2 | 77.7 |
> |**Generation-based Methods**|-|-|-|-|-|
> | Img2Loc(LLaVA)     | 7.9| 14.2| 19.5 | 29.9| 39.7|
> | G3(LLaVA)          | 23.9| 33.6 | 44.0| 60.6| 75.9 |
> | GaGA               | 6.9  | 18.9 | 34.5 | 56.7 | 71.6 |
> | GaGA(MP16-Pro)       | 24.3 | 33.7 | 43.9 | 61.4 | 76.3 |
>
> While our base `GaGA` model is competitive with other generation-based methods, the key insight comes from our augmented model, `GaGA(MP16-Pro)`, which shows a significant performance leap. On Im2GPS3k, it surpasses all listed traditional and generative methods at the crucial fine-grained levels (15.0% @1km and 37.1% @25km). The improvement is even more pronounced on YFCC4k, where `GaGA(MP16-Pro)` achieves a top-tier score of 24.3% @1km and 33.7% @ 25km.
>
> We believe **Table C** and **Table D** effectively demonstrate GaGA's strong performance on standard benchmarks and show its potential when integrated with retrieval-based tools. We will add these experiments to our revised manuscript. Thank you again for pushing us in this valuable direction.
>
> ---
> **Inconsistency Between Granularity and Performance (Limitations Weaknesses #2)**
>
> The "finer granularity" of our MG-Geo dataset refers not to a denser sampling of coordinates, but to the richness of the **reasoning data**—the multi-step clues and dialogues—which is designed to facilitate a fundamental shift from "coordinate matching" to "scene understanding and reasoning."
>
> This focus on reasoning explains the performance profile you observed. As we also discussed in our response to **Reviewer Ftex (Performance Gap Explanation)**, unlike retrieval or classification-based models optimized for matching visual patterns, our generative approach undertakes the more complex task of generating coordinates from an understanding of the scene. While this currently faces challenges with numerical precision, it enables GaGA to excel where other models cannot. Specifically, it achieves state-of-the-art performance on **semantic localization** (predicting country and city names) and obtains a superior **Geoscore**, which indicates a more plausible and contextually aware geographic prediction, even if the point-wise accuracy is lower.
>
> We believe the most promising path forward is not just adding denser cues, but enhancing the model's capabilities. As demonstrated in our new experiments on the YFCC4k and Im2GPS3k benchmarks, when we augment GaGA with a RAG-based retrieval tool, its fine-grained localization performance improves dramatically. These examples show that our core contribution—a model capable of reasoning—can be powerfully combined with external tools to achieve both state-of-the-art precision and explainability, paving the way for the next generation of geolocation models. Furthermore, building on this "tool-use" philosophy, we are already undertaking new research that integrates an external, diffusion-based tool to specifically address the challenge MLLMs face in generating precise latitude and longitude coordinates.
>
> [^1] Astruc, G., Dufour, N., Siglidis, I., Aronssohn, C., Bouia, N., Fu, S., ... & Landrieu, L. (2024). Openstreetview-5m: The many roads to global visual geolocation. In _Proceedings of the IEEE/CVF Conference on Computer Vision and Pattern Recognition_ (pp. 21967-21977).
>
> [^2] Clark, B., Kerrigan, A., Kulkarni, P. P., Cepeda, V. V., & Shah, M. (2023). Where we are and what we're looking at: Query based worldwide image geo-localization using hierarchies and scenes. In _Proceedings of the IEEE/CVF Conference on Computer Vision and Pattern Recognition_ (pp. 23182-23190).

---

> > ### Comment · Reviewer_UGjh · 2025-08-04
> > **Response to author**
> >
> > Well, the author responded to some of my questions. I hope these updated experimental results can be presented in the final version. I decided to raise the score by one point to weekly accept.

---

> > > ### Author Response · Authors · 2025-08-04
> > >
> > > Dear Reviewer UGjh,
> > >
> > > Thank you very much for your insightful review and valuable feedback on our paper. We commit to incorporate the results and corresponding analysis into the final version of the paper.
> > > Thank you again for your time and expertise.
> > >
> > > Sincerely,
> > >
> > > The Authors

---

### Official Review · Reviewer_Ftex · 2025-07-01

**Rating:** 5
**Confidence:** 4

**Summary:**

This paper introduces MG-Geo, the first large-scale multimodal dataset for global geolocation, comprising 5 million geographically diverse dialogues across 210 countries with fine-grained cues. Leveraging this dataset, the authors develop GaGA, a novel MLLM specifically designed for geolocation that outperforms SOTA baselines in administrative boundary prediction and exhibits interactive refinement capabilities.

**Dataset Code Accessibility:**

Yes

**Ethical Considerations:**

No, there are no or only very minor ethics concerns

**Final Justification:**

The authors' response has addressed my concerns. Given the detailed experiments and the positive results, I am willing to raise the rating to accept.

**Limitations Weaknesses:**

1. This paper lacks ablation studies on the components of MG-Geo dataset (Meta/Clue/Dialog). If such experiments were conducted, it would help to better understand the superiority of MG-Geo dataset over other datasets.
2. GaGA's performance in coordinate prediction seems to be inferior to that of PIGEON and OSV-5M-Baseline, due to the inherent limitations of conventional LLMs.
3. Figure 3 shows that GaGA struggles with confusing similar architectural styles in Europe. Could the authors provide an analysis of this failure case?
4. There seems to be a writing error. It might be better to change "architectural" to its noun form in the top left corner of Figure 2.

**Strengths Contributions:**

1. The writing has a clear structure.
2. The construction of MG-Geo dataset is innovative due to multimodal fusion and automated annotation.
3. The model GaGA, trained on MG-Geo dataset, outperforms SOTA baselines in administrative boundary prediction. Especially, GaGA achieves 63.06% country-level accuracy, surpassing OSV-5M-Baseline by 4.57%.

---

> ### Author Rebuttal · Authors · 2025-07-30
>
> **Ablation Study Clarification (Limitations Weaknesses #1)**
>
> Thank you for your careful review. Below are the ablation studies conducted on the components of the MG-Geo dataset.
>
> Our goal with this study is not merely to optimize for a single metric, but to show how each data component helps construct a model that achieves both high-precision localization and strong capabilities in interpretability and dialogue.
>
> **Experimental Setup**
> Using the test set from Section 5.3, we design five core experimental groups for a thorough comparison. To ensure a fair evaluation, the model architecture, hyperparameters, and training strategy are held constant across all experiments. The only variable is the composition of the training data. The performance is evaluated based on four key metrics, which are consistent with Table 7 in Appendix D.
>
> **Evaluation Metrics**
> The metrics for this ablation study are defined as follows:
> -   *Accuracy:* The Geoscore, where a higher value indicates a smaller average prediction error.
> -   *Informativeness:* Assessed by an LLM Judge, this measures the hit rate of the model's generated clues against the ground-truth clues.
> -   *Fluency:* Scored by an LLM Judge on a Likert scale to evaluate the conversational coherence and naturalness of the generated text.
> -   *Relevance:* Evaluated by an MLLM Judge to assess the contextual relevance between the generated text and the input image.
>
> **Table A**: Ablation Studies on Components of MG-Geo Dataset
> |Training Data|Accuracy|Informativeness|Fluency|Relevance|
> |-|-|-|-|-|
> |Meta|3872|0.6164|4.5722|2.2944|
> |Meta+Clue|3835|0.6172|4.4944|2.1685|
> |Meta+Dialog|3854|0.7549|4.9296|3.5796|
> |Meta+Clue+Dialog|3817|0.7809|4.9555|3.6506|
> |Clue+Dialog|3773|0.7674|4.9259|3.8981|
>
> **Analysis**
> Our ablation study, detailed in **Table A**, reveals the distinct and synergistic roles of the `Meta`, `Dialog`, and `Clue`components in the MG-Geo dataset.
>
> The results establish the primary functions of our two main data components. The `Meta` data is the bedrock of localization accuracy; the model trained on `Meta` alone achieves the highest **Accuracy** (3872), while the configuration without it (`Clue`+`Dialog`) has the lowest. Conversely, the `Dialog` data is the core driver of reasoning and conversational ability. Adding `Dialog` to `Meta` data results in a dramatic improvement across all interactive metrics, including a significant jump in **Informativeness** (from 0.6164 to 0.7549).
>
> The `Clue` data plays a more nuanced, conditional role. When added to `Meta` data alone, the fragmented clues slightly interfere with the model's pure localization task. However, when added to a model already trained on the `Dialog`reasoning framework (`Meta`+`Clue`+`Dialog`), the `Clue` data acts as a powerful enhancer. It injects targeted factual knowledge, pushing **Informativeness** and **Fluency** to their highest levels (0.7809 and 4.9555, respectively). This demonstrates a clear synergy where `Meta` builds the foundation, `Dialog` creates the reasoning structure, and `Clue` refines that reasoning with key knowledge.
>
> Finally, the study confirms an intentional design trade-off. While the `Meta`-only model is the most precise in static localization, incorporating `Dialog` and `Clue` data creates a more capable and interactive model. This slight trade-in precision for a massive gain in reasoning and conversational skill is central to our work's goal: to create a balanced, interpretable geolocation assistant, a feature not facilitated by other existing datasets.
>
> In summary, our ablation study successfully validates the deliberate design of the MG-Geo dataset. Each component has a distinct and indispensable role in creating a model that effectively balances high-precision localization with deep, explainable reasoning and interaction. We will incorporate these findings into our revised manuscript.
>
> ---
> **Performance Gap Explanation (Limitations Weaknesses #2)**
>
> Thank you for this very important observation. You are correct that there is a performance difference in raw coordinate prediction. We believe this stems from a fundamental distinction between our generative approach and the dominant paradigms in the field, which are typically classification-based, retrieval-based, or hybrid methods that combine both.
>
> While these classification and retrieval methods achieve strong performance on established benchmarks, they face inherent limitations. Classification-based approaches can suffer from significant errors when an image's true location is far from the center of its correctly predicted grid cell. Retrieval-based methods depend on massive, hard-to-maintain image databases, and their performance degrades sharply on out-of-distribution samples lacking a close visual match. Crucially, both paradigms often act as "black boxes." They lack interpretability and a deep, generalizable understanding of geography, as their success relies on visual pattern matching rather than genuine reasoning.
>
> In contrast, GaGA treats geolocation as an end-to-end generation task, forcing the model to genuinely "understand" the visual information (architecture, vegetation, text, etc.) to generate coordinates. This approach, while aiming for a more fundamental and generalizable world understanding, faces its own technical challenges, as LLMs are not inherently optimized for generating precise, long floating-point numbers [^1].
>
> Despite this, our paradigm offers unique and significant value. The core contribution of our work is to shift the field from **"matching coordinates"** toward **"understanding a scene and explaining the reasoning."** Our approach demonstrates a superior capability in "semantic localization," achieving state-of-the-art performance in predicting country and city names. Furthermore, on the **Geoscore** metric—which measures the average error level between the prediction and the true value—our model surpasses existing methods.
>
> We acknowledge the current precision limitations of a purely generative method. However, we see this not as a dead end, but as a starting point. The future of this approach is promising. For instance, the powerful **tool-use capabilities** of MLLMs could be leveraged to overcome this. A model like GaGA could first reason about the general location and then call a specialized tool to calculate the precise coordinates, combining the strengths of both worlds. For experimental results supporting this direction, we would also refer you to our response to **Reviewer UGjh (Limited Benchmark Comparisons and Incomplete Baseline Comparison)** .
>
> ---
> **Analysis of Failure Case (Limitations Weaknesses #3)**
>
> Thank you for this insightful question and for prompting this deeper analysis. We have analyzed this behavior and conclude that it is a phenomenon we term the **"Similarity Trap,"** which arises from the inherent **ambiguity** of many geographical clues—a core challenge in geolocation [^2].
>
> This is a fundamental problem that also affects traditional methods. For instance, a retrieval-based model might fetch images from multiple European cities with similar architecture, while a classification model might struggle to distinguish between geo cells in different countries that share a similar visual style. Crucially, once these static models fall into the trap, they offer no mechanism for correction.
>
> This is where GaGA's interactive design offers a unique advantage. Unlike static models, GaGA can be guided out of the "Similarity Trap" through human intervention. As demonstrated in Figure 3, the model is initially misled by ambiguous European features. However, when the user provides a more specific clue (e.g., specifying "Northern Europe"), the model dynamically adjusts its reasoning and successfully converges on the correct location in Denmark. This highlights GaGA's ability to leverage new information to correct its path.
>
> To empirically validate the impact of ambiguity of geographical clues, we design an experiment evaluating performance based on the specificity of the guiding clues. We partition the test set from Section 5.3 into two distinct subsets:
> -   **High-Specificity Subset:** Contains questions about geographically unique features. (e.g., _"What language is on the sign?"_, _"Which side of the road are vehicles on?"_)
> -   **Ambiguous Subset:** Contains questions about features that are geographically widespread or vague. (e.g., _"Describe the architectural style."_, _"What is the natural landscape?"_)
>
> **Table B**: Impact of Ambiguity
> |Subset Type|Country Accuracy|Region Accuracy|City Accuracy|Accuracy @1km|Accuracy @25km|Accuracy @200km|Accuracy @750km|Accuracy @2500km|
> |-|-|-|-|-|-|-|-|-|
> |Ambiguous Subset|59.5|26.2|7.0|4.1|16.6|41.2|75.8|92.5|
> |High-Specificity Subset|62.2|28.9|9.1|5.8|19.2|41.0|80.7|94.4|
>
> As shown in Table B, the model achieves significantly **better performance on the High-Specificity Subset** and **poorer performance on the Ambiguous Subset** . This result supports our analysis that performance is heavily influenced by the ambiguity of the guiding clues, and it underscores the importance of the interactive, corrective dialogue process that is central to GaGA's design. We will add this analysis to the paper.
>
> ---
> **Writing Error (Limitations Weaknesses #4)**
>
> Thank you for your careful proofreading and attention to detail. We have changed "architectural" to its noun form, "architecture," in Figure 2 in our revised manuscript. We appreciate you pointing this out.
>
> [^1]: Singh, A. K., & Strouse, D. J. (2024). Tokenization counts: the impact of tokenization on arithmetic in frontier llms. _arXiv preprint arXiv:2402.14903_.
>
> [^2]: Dufour, N., Kalogeiton, V., Picard, D., & Landrieu, L. (2025). Around the world in 80 timesteps: A generative approach to global visual geolocation. In _Proceedings of the Computer Vision and Pattern Recognition Conference_ (pp. 23016-23026).

---

> ### Comment · Reviewer_Ftex · 2025-08-05
> **Response to Authors**
>
> The authors' response has addressed my concerns. Given the detailed experiments and the positive results, I am willing to raise the rating to accept.

---

> > ### Author Response · Authors · 2025-08-05
> >
> > Dear Reviewer Ftex,
> >
> > Thank you very much for your insightful review and valuable feedback on our paper. We agree that the additional experiments you suggested are important and will significantly strengthen our work. We commit to incorporate the results and corresponding analysis into the final version of the paper.
> >
> > Thank you again for your time and expertise.
> >
> > Sincerely,
> >
> > The Authors

---

### Official Review · Reviewer_sVAq · 2025-07-03

**Rating:** 4
**Confidence:** 2

**Summary:**

In this paper, the authors propose a high-quality multimodal global geolocation dataset called MG-Geo, and a new multimodal large language model GaGA developed based on this dataset. MG-Geo contains five million instances of geographic conversation data from 210 countries, covering rich geographic clues such as road markings, vegetation, language.

**Dataset Code Accessibility:**

Yes

**Ethical Considerations:**

No, there are no or only very minor ethics concerns

**Final Justification:**

The author's rebuttal addresses my concerns. I agree with other reviewers and decide to raise the origin score.

**Limitations Weaknesses:**

I have the following questions about this paper:
1) How is the proposed dialogue dataset used during fine-tuning?
2) How does GaGA perform if the QA is used only during inference, without being included in the training phase?
3) I conducted a simple test by feeding the two images from Figure 3 into ChatGPT. After a few rounds of dialogue, it produced the results {"latitude": 51.5185, "longitude": -0.1439} and {"latitude": 55.68, "longitude": 12.57}, respectively—closely matching the outputs generated by GaGA. What are the specific advantages of GaGA compared with ChatGPT? More critically, ChatGPT achieves this without any training on the MG-Geo dataset.

**Strengths Contributions:**

1) This paper introduces a large-scale, high-quality multimodal dialogue dataset;
2) The proposed GaGA can conduct multiple rounds of dialogue with users, gradually narrowing the location range based on user prompts and questions.

---

> ### Author Rebuttal · Authors · 2025-07-30
>
> We thank the reviewers for their constructive feedback and are encouraged by the consensus on our work's primary contributions. The reviewers broadly recognized **MG-Geo** as the first large-scale, high-quality, multimodal dataset for global geolocation, and highlighted the **GaGA** model for its state-of-the-art performance, interactivity, and interpretability.
>
> **Data Utilization Clarification (Limitations Weaknesses #1)**
>
> Thank you for your meticulous review. Regarding your question about the utilization of the dialogue dataset during fine-tuning, our approach is as follows: The training process comprises two distinct stages, and the "Dialog part" of MG-Geo is applied exclusively in the second stage.
>
> The purpose of fine-tuning with the "Dialog part" is to teach GaGA a structured reasoning methodology. Instead of merely associating an image with a final answer, the model learns to deconstruct the complex geolocation problem into a series of manageable, sequential steps. This approach, inspired by Chain-of-Thought (CoT), also equips the model with the crucial ability to integrate corrective feedback, thereby enhancing the interpretability and adaptability of its reasoning process.
>
> In each sample of "Dialog part", a "questioner" prompts the "geo-guesser" to analyze various clues—such as environment, climate, architecture, and cultural elements—to progressively narrow down the location. Each conversational turn represents a step in a logical chain, which ultimately leads to a location prediction. This structured dialogue format effectively guides the model's reasoning path to the final decision.
>
> **Performance Without Training QA (Limitations Weaknesses #2)**:
>
> Regarding the performance of GaGA when the QA dialogue is used only during inference and not for training, we have conducted a specific experiment to address this.
>
> As detailed in **Table 10** of our paper, we compared the performance of GaGA on the GWS15k benchmark under two conditions:
>
> 1.  *GaGA-pretraining:* This version was trained only on the large-scale metadata and was not fine-tuned on any dialogue data.
>
> 2.  *GaGA-finetuning*: This is our final model, which was fully fine-tuned on the complete dataset, including the dialogue component.
>
> **Table 10**: Impact of Training Framework on GaGA’s performance
> |Method|Country|Region|City|
> |-|-|-|-|
> |LLaVA-Llama3|1.76|0.26|0.02|
> |GaGA-pretraining|63.38|28.84|6.47|
> |GaGA-finetuning|63.06|27.95|6.28|
>
> Our findings show that the `GaGA-pretraining` model achieves slightly higher raw localization accuracy. However, the final `GaGA-finetuning` model, by training on the dialogue data, acquires the ability to generate much more fluent, interpretable, and interactive reasoning steps. We believe this trade-off—a minor, acceptable decrease in localization performance for a significant gain in interpretability and conversational capability—is a valuable one.
>
> For a more detailed ablation study on the contribution of each component of MG-Geo, we would also kindly refer you to our response to **Reviewer Ftex (Ablation Study Clarification)**, where we have provided further experimental evidence.
>
> **Advantage Over ChatGPT (Limitations Weaknesses #3)**
>
> We appreciate you taking the time to test the images with ChatGPT, and we agree that its capabilities are impressive.
>
> The key difference is that our work provides a complete, open-source ecosystem (GaGA model, MG-Geo dataset, GWS15k benchmark), whereas ChatGPT is a closed-source "black box." This open approach is crucial for scientific progress, as it allows the community to verify our results, build upon our work, and advance the field. A proprietary model, regardless of its capabilities, cannot serve as a reproducible scientific artifact.
>
> What's more, ChatGPT's strong performance is not surprising, as its massive, proprietary training data almost certainly contains vast amounts of relevant geotagged images. For a fairer, apples-to-apples comparison, GaGA significantly outperforms other open-source MLLMs of a similar scale (as shown in **Table 6** in our paper). And as a specialized model, GaGA is far more cost-effective, scalable, and private for large-scale, real-world applications compared to relying on expensive API calls to a massive, general-purpose model.
>
> **Table 6**:Performance of Advanced MLLMs with Different Types of Prompt Inputs
> |Method|Country|Region|City|
> |-|-|-|-|
> |GaGA|64.89|27.97|7.67|
> |Qwen-VL|13.89|6.03|2.01|
> |InternVL2|54.11|19.19|3.29|

---

> ### Comment · Area_Chair_z8e6 · 2025-08-01
> **discussion phase starts**
>
> Dear Reviewers,
>
> Given the diverse ratings this paper received, and considering the authors' detailed responses to each review, we would appreciate your re-evaluation of both the reviews and rebuttal to provide your revised assessment in the next days.
>
> Best regards,
>
> Your AC

---

### Note · Authors · 2025-08-12

Dear ACs, SACs, and PCs,

We would like to sincerely thank the reviewers, ACs, SACs and PCs for their time, effort, and constructive feedback, which has helped us substantially improve the quality of our manuscript. Our work presents the following significant contributions to the field of global geolocation:

**A Novel Dialogue-based Geolocation Dataset (MG-Geo):** We introduce MG-Geo, the first large-scale, multi-modal dataset specifically designed for global geolocation through interactive dialogue—**a contribution recognized by all reviewers**. Comprising rich metadata, targeted visual clues, and multi-step reasoning dialogues, MG-Geo moves beyond simple image-coordinate pairs. It is explicitly designed to teach models the process of geographic problem-solving—how to analyze evidence, form hypotheses, and refine conclusions—a critical aspect missing from existing datasets.

**A New Paradigm for Interactive Geolocation:** We propose GaGA, a novel generative model that pioneers a new paradigm for geolocation. Instead of treating the task as a static, "black-box" prediction, GaGA frames it as an interactive and interpretable reasoning process. Our model can articulate its rationale step-by-step, engage in dialogue to analyze visual evidence, and dynamically correct its predictions based on new information, more closely emulating human-like problem-solving. **This contribution has been acknowledged by Reviewers sVAq, UGjh and fYB5.**

**Rigorous Validation of Dataset Design:** Following the suggestions of **Reviewer Ftex**, we conduct comprehensive ablation studies that rigorously validate the design of our MG-Geo dataset. These experiments systematically demonstrate the distinct and synergistic roles of each data component.

**State-of-the-Art Performance on Standard Benchmarks:** **Table C, D and Table 2 in our paper**  demonstrate that GaGA achieves highly competitive and SOTA performance. On its own, GaGA excels at semantic localization (e.g., country/city prediction). Furthermore, when enhanced with a standard RAG tool, GaGA achieves top-tier accuracy on challenging benchmarks like Im2GPS3k and YFCC4k, proving that our reasoning-first approach provides a powerful foundation for achieving both high precision and interpretability.

Best.

---

### Decision · Program_Chairs · 2025-09-18

**Decision:**

Reject

**Comment:**

This paper presents a multimodal global geolocation dataset that incorporates fine-grained geographic element cues and multi-turn reasoning dialogues. Building on this dataset, the authors propose a multimodal large language model trained through a two-stage pipeline. The paper received evaluations from four reviewers, yielding one "Accept" and three "Borderline Accept" ratings—all positive. Following a comprehensive assessment of the manuscript, the reviews, and the authors' rebuttal, the area chairs have decided to accept the paper.

===== FINAL UPDATE FROM DB Track PCs ====

The final decision for this paper has been taken by the program chairs after consultation with the SACs. All Senior Area Chairs have ranked papers according to the feedback from the AC during the review process. We decided to leave the original meta-review to reflect the opinion of the AC in light of the initial discussions with reviewers and SAC.